https://doi.org/10.1038/s43856-022-00223-3　　OPEN
# Intraglandular mesenchymal stem cell treatment induces changes in the salivary proteome of irradiated patients

Charlotte Duch Lynggaard [1,2 ✉], Rosa Jersie-Christensen[3], Morten Juhl [4], Siri Beier Jensen [5], Christian Grønhøj[1], Jacob Melchiors[1], Søren Jacobsen[2,6], Michael Møller-Hansen[7], Mikkel Herly [8], Annette Ekblond[4], Jens Kastrup[4], Anne Fischer-Nielsen[9], Daniel Belstrøm [10,11] & Christian von Buchwald[1,2,11]

## Abstract

**Background** Hyposalivation and xerostomia (dry mouth), are the leading site-effects to treatment of head and neck cancer. Currently, there are no effective therapies to alleviate radiation-induced hyposalivation. Adipose tissue-derived mesenchymal stem/stromal cells (AT-MSCs) have shown potential for restoring salivary gland function. However, the mode of action is unknown. The purpose of the present study was therefore to characterize the effect of AT-MSC therapy on the salivary proteome in previously irradiated head and neck cancer patients.

**Methods** Whole saliva was collected from patients with radiation-induced salivary gland hypofunction ($n = 8$) at baseline, and 120 days after AT-MSC treatment, and from healthy controls ($n = 10$). The salivary proteome was characterized with mass spectrometry based proteomics, and data was compared within the AT-MSC group (baseline versus day 120) and between AT-MSC group and healthy controls. Significance levels between groups were determined by using double-sided t-test, and visualized by means of principal component analysis, volcano plots and cluster analysis.

**Results** Here we show that 140 human proteins are significantly differentially expressed in saliva from patients with radiation-induced hypofunction versus healthy controls. AT-MSC treatment induce a significant impact on the salivary proteome, as 99 proteins are differentially expressed at baseline vs. 120 days after treatment. However, AT-MSC treatment does not restore healthy conditions, as 212 proteins are significantly differentially expressed in saliva 120 days after AT-MSCs treatment, as compared to healthy controls.

**Conclusion** The results indicate an increase in proteins related to tissue regeneration in AT-MSCs treated patients. Our study demonstrates the impact of AT-MSCs on the salivary proteome, thereby providing insight into the potential mode of action of this novel treatment approach.

## Plain language summary

Currently, there are no effective treatments to ease dry mouth, which is a leading long-term side effect of radiation treatment for head and neck cancer. However, treatment with stem cells has shown potential for restoring function of the salivary glands, which are damaged due to radiation. We compared proteins in saliva of previously radiation-treated patients with healthy non-irradiated persons and found differences in the levels of 140 proteins. After stem cell treatment of irradiated patients, we found changes in the salivary content of proteins related to tissue regeneration. Our study demonstrates the impact of stem cell treatment on proteins in saliva, thereby providing insight into the potential mode of action of this treatment approach for patients with radiation-induced dry mouth. Consequently, this could potentially help to improve treatment of dry mouth in the future.

[1] Department of Otolaryngology, Head and Neck Surgery and Audiology, Rigshospitalet, Copenhagen University Hospital, Copenhagen, Denmark. [2] Department of Clinical Medicine, Copenhagen University, Copenhagen, Denmark. [3] Department of Science and Environment, Roskilde University, Roskilde, Denmark. [4] Cardiology Stem Cell Centre, Rigshospitalet, Copenhagen University Hospital, Copenhagen, Denmark. [5] Department of Dentistry and Oral Health, Aarhus University, Aarhus, Denmark. [6] Copenhagen Research Center for Autoimmune Connective Tissue Diseases - COPEACT, Rigshospitalet, Copenhagen University Hospital, Copenhagen, Denmark. [7] Department of Ophthalmology, Rigshospitalet, Copenhagen University Hospital, Copenhagen, Denmark. [8] Department of Plastic Surgery and Burns Treatment, Rigshospitalet, Copenhagen University Hospital, Copenhagen, Denmark. [9] Department of Clinical Immunology, Rigshospitalet, Copenhagen University Hospital, Copenhagen, Denmark. [10] Department of Odontology, Section for Clinical Oral Microbiology, Faculty of Health and Medical Sciences, University of Copenhagen, Copenhagen, Denmark. [11] These authors contributed equally: Daniel Belstrøm, Christian von Buchwald. ✉email: clynggaard@dadlnet.dk

Approximately 70–80% of patients with head and neck cancer undergo radiation therapy, either as a single therapy or in combination with chemotherapy and surgery[1]. Despite substantial improvements to decrease side effects after radiation therapy using hypofractionated and intensity-modulated radiation therapy (IMRT), the salivary glands are typically damaged by ionizing radiation[2].

Radiation damage to the salivary glands induces a complex cascade of acute and late responses, which are not fully understood. However, there is a consensus that it promotes chronic changes characterized by inflammation and interstitial fibrosis, concomitant with loss of acinar cells, local stem/progenitor cells, and blood vessels, thereby decreasing the production and quality of saliva[2–4].

Saliva is critical for oral homeostasis, speech, mastication and swallowing. Indeed, salivary gland hypofunction induced by radiation therapy often results in dry mouth (xerostomi)a, increased risk of oral infections, and impaired speech, taste sensation, chewing and swallowing, oral mucosal discomfort, and a worsened nutritional state. Further, patients report pain and trouble sleeping, which has profound impact on their quality of life[5,6]. Finally, the reduced salivary flow rate may lead to dental decay with impairment of oral function that often leads to expensive dental treatments, which adds to social inequality among cancer survivors. Currently, the treatment for salivary gland hypofunction is limited to artificial saliva, which has a limited lubricating effect of short duration and lacks the protective effects of saliva or therapies attempting to stimulate the residual capacity of the salivary glands[7]. Hence, there is a need for new treatments.

Mesenchymal stem/stromal cells (MSCs) can be isolated from most vascularized tissues, and following ex vivo expansion, they can be cryopreserved to offer an off-the-shelf treatment option for clinical use. Although the mode of action is not fully understood, in vitro and animal studies indicate that they are not likely to engraft, but they probably possess supportive and paracrine functions, such as anti-apoptosis, regeneration, immunomodulation, angiogenesis, and anti-scarring, as well as support the growth and differentiation of endogenous stem and progenitor cells[8–12]. MSC therapy is being investigated in multiple clinical trials for diverse disorders such as graft-versus-host-disease, heart disease, and Crohn's disease[13–17]. Recently, our group published findings from a randomized controlled trial showing promising results after treatment with autologous adipose tissue-derived mesenchymal stem/stromal cells (AT-MSCs) of the submandibular glands in patients treated for an oropharyngeal squamous cell carcinoma, including increased unstimulated salivary flow rate[18,19]. Whereas our initial study focused on autologous AT-MSCs, the present study investigates treatment with allogeneic cells from healthy donors. As a benefit, the patients themselves avoid liposuction to obtain AT-MSCs for culture expansion, and the cell product is thoroughly characterized and standardized.

Mass spectrometry analysis provides detailed information on the human proteome. Further, it offers an opportunity to study the proteomes of cohabiting microbial origin (e.g., in saliva) from a single clinical sample[20,21]. Previous reports have demonstrated that this approach can be utilized to detect differences in patients with oral disease compared to healthy controls[22–24]. Therefore, we hypothesize that proteome analysis of the whole saliva can provide new knowledge not only about the regenerative processes induced by AT-MSCs to the salivary glands but also if these affects the oral microbiome. Accordingly, this study had two aims: first, to cross-sectionally compare the salivary proteome in patients with radiation-induced salivary gland hypofunction with age- and sex-matched healthy controls and second, to characterize longitudinally the impact of intraglandular treatment with allogeneic AT-MSCs on the salivary proteome, in patients with radiation-induced hypofunction during a 120-day clinical trial (Fig. 1).

The main finding from the present study was that treatment with locally injected allogeneic AT-MSCs induced marked changes to the salivary proteome, and that effects of the treatment were persistent after 120 days, as evaluated by the composition of the salivary proteome. However, AT-MSC treatment did not restore the composition of the salivary proteome to healthy conditions, as the salivary proteome in irradiated patients 120 after AT-MSC treatment remained significantly different from that of healthy controls.

## Methods

**Study design.** Details on the study design, population and intervention has already been published[25]. This study was an investigator-investigator-initiated, open-label, non-randomized, first-in-man, single-site study designed to evaluate the safety and efficacy of using allogeneic AT-MSCs to treat radiation-induced hyposalivation. The patients were examined at follow-up visits on day 1, 5, 30, and 120 to assess potential adverse events and to collect saliva. We evaluated possible changes in the salivary proteome in samples from day 0 (intervention), day 5 and day 120. The study was monitored by the Good Clinical Practice (GCP) Unit of Copenhagen and approved by the Danish Medicines Agency (Eudra-CT 2018-003856-19), the National Committee on Health Research Ethics (H-1808924) and the Danish Data Protection Agency (VD-2018-476, I-6735). The study was registered at clinicaltrials.gov (number NCT03874572), in which composition of the salivary proteome was recorded as a pre-specified secondary endpoint. The study was conducted according to protocol and complied with the Declaration of Helsinki, and written informed consent was obtained from all participants before enrolment.

**Study population.** The main inclusion criteria were previous radiation therapy of oropharyngeal squamous cell carcinoma stage I–II, Union for International Cancer Control version 8 (UICC-8), subjective and objective signs of salivary gland hypofunction and minimum of 2 years without recurrence after radiation therapy. The main exclusion criteria were cancer within the last 4 years, xerogenic medications, allergy to penicillin or streptomycin, other diseases of the salivary glands and any previous stem cell therapy.

The dental status of all study participants (both patients and healthy controls) was assessed using orthopantomography, and we received a copy of their last dental record (no more than 12 months old). Due to COVID-19, dental examinations were not performed. The dental records were evaluated by an experienced dentist.

The study population has been described in detail[25]. In brief, the patient group was comprised of ten patients (7 men and 3 women, 59.5 (range: 45–70) years, with a median of 5.5 years after radiation therapy), all with clinically evident salivary gland hypofunction (as assessed by sialometry) and xerostomia following radiation therapy for an p16+ oropharyngeal squamous cell carcinoma. The patients had previously received photon therapy to a full dosage to the tumor and lymph node metastases of 66–68 Gray delivered in 2 Gray per fraction with six fractions per week concurrent with cisplatin therapy. This radiation therapy delivers practically the whole prescribed radiation dose to the ipsilateral submandibular gland and a minor fraction dose to the lower portion of the ipsilateral parotid gland. The control group was comprised of ten healthy participants who were

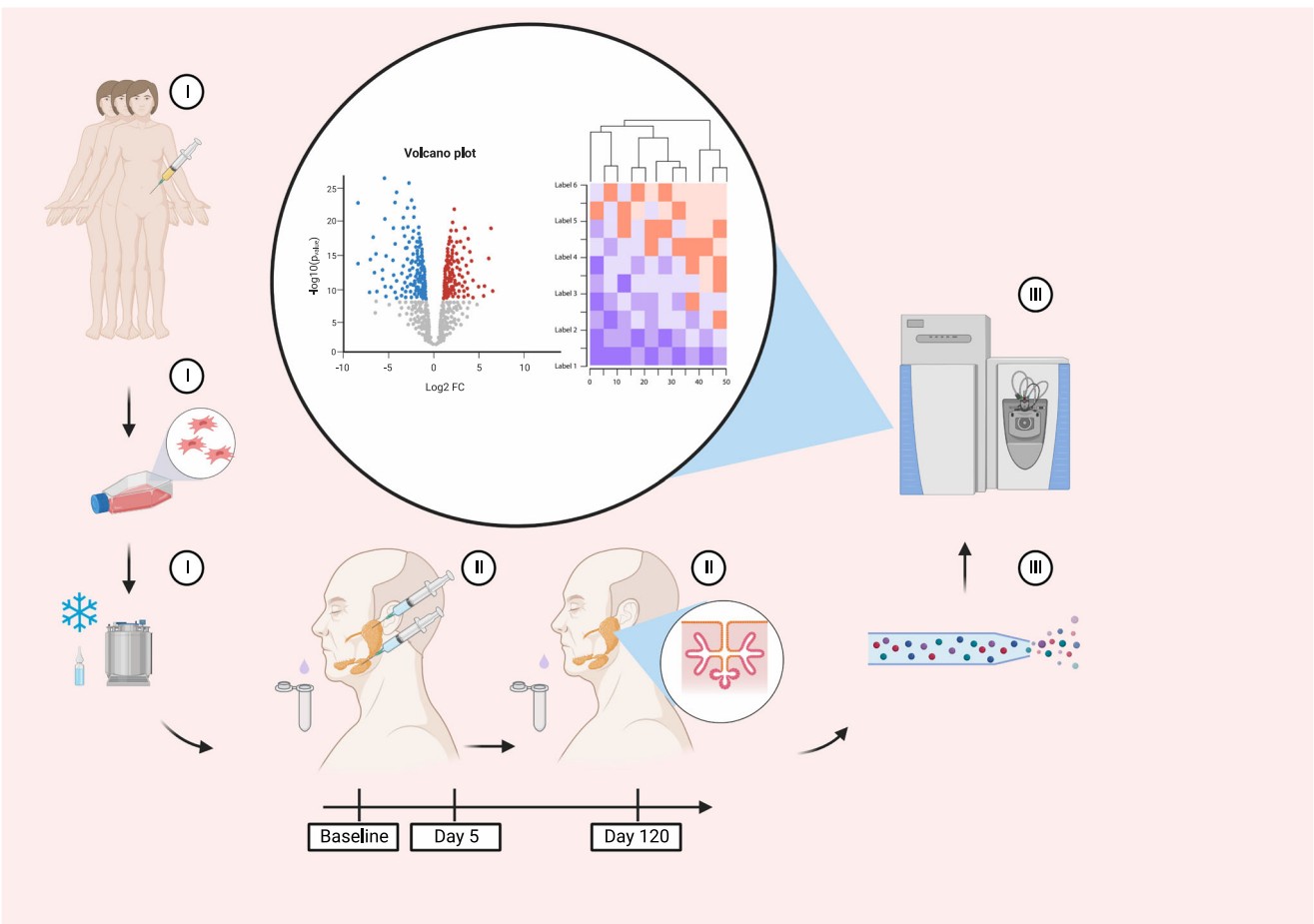

**Fig. 1 Schematic depiction illustrating the analyses of stimulated whole saliva from patients treated with mesenchymal stem/stromal cells.** Adipose tissue-derived mesenchymal stem/stromal cells (AT-MCSs) from three healthy female donors were used as a study intervention (I). AT-MSCs were injected in the parotid and submandibular glands in ten patients with radiation-induced salivary gland hypofunction and xerostomia and stimulated whole saliva was collected by sialometry (II). Whole-saliva samples from baseline (prior to the AT-MSC treatment), day 5 and 120 days after intervention were analyzed by nanoscale liquid chromatography-tandem mass spectrometry to explore changes induced by the AT-MSCs (III). Created using Biorender (VR241Q1V5K).

matched with the patient group according to age, sex, ethnicity, and education level. Of the included patients, two failed to produce sufficient saliva for analysis. Hence, the analyses are based on saliva from eight patients and ten control persons.

**The AT-MSC intervention.** The AT-MSC product was manufactured by Cardiology Stem Cell Centre (CSCC), University Hospital Copenhagen, under a Manufacturing and Importation Authorization granted by the Danish Medicines Agency. Lipoaspirates were obtained from three healthy consenting female donors (22–26 years old) according to an established protocol[26–28]. Cells were isolated from lipoaspirates by enzymatic digestion with collagenase, and the AT-MSCs were expanded for two passages in automated closed bioreactor systems (Quantum Cell Expansion System, Terumo BCT) with 5% human platelet lysate as the growth supplement (Sexton Biotechnologies) in MEM alpha (Gibco) and Penicillin/Streptomycin (Gibco). The AT-MSCs were cryopreserved at $50 \times 10^6$/mL CryoStor (BiolifeSolutions) in CellSeal vials (Sexton Biotechnologies) and stored at $<-180\,°C$ in nitrogen dry-storage until clinical use. Authorization of tissue establishment for the handling of human tissues and cells has been licensed by The Danish Patient Safety Authority. The test group received ultrasound-guided injections of $25 \times 10^6$ AT-MSCs bilaterally in the submandibular glands and $50 \times 10^6$ AT-MSCs in each parotid

gland between April 2020 and January 2021 at Rigshospitalet, University Hospital of Copenhagen, Denmark.

**Saliva collection.** Saliva was collected by sialometry under the same conditions for patients and controls[29,30]. The participants were requested to drink a minimum of 2 L water the day before saliva collection by sialometry and were to refrain from eating, drinking and oral manipulation a minimum of 1 h prior to the collection of saliva. The saliva collection was performed in the same room between 10:00 a.m. and 12:00 noon. Following the collection of unstimulated whole saliva (which was used to determine unstimulated salivary flow rate and hypofunction) for 10 min, the patients received a tasteless paraffin pellet (Ivoclar Vivadent©). After 1 min of chewing the first pellet, the stimulated saliva was swallowed after which stimulated whole saliva was collected for 5 min and weighed. The saliva samples were stored in Cryotubes and snap frozen in liquid nitrogen and stored at −80 °C until analyses.

**Sample preparation.** In all, 500 µL stimulated whole saliva was lysed with 500 µL 95 °C warm lysis buffer (6 M Guanidinium hydrochloride (GuHCl), 5 mM tris(2-carboxyethyl)phosphine (TCEP), 10 mM chloro-acetamide (CAA), 100 mM Tris–HCl pH 8.5), followed by additional heating at 95 °C for 10 min. The lysates were sonicated with micro tip probe (Sonic materials

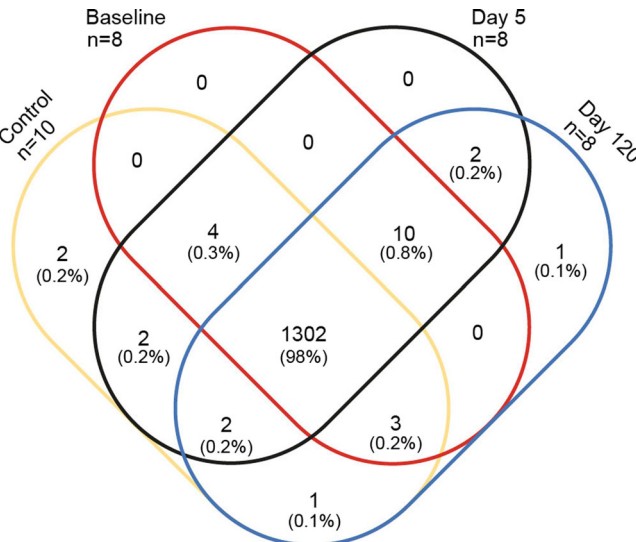

**Fig. 2 Venn diagram of identified proteins.** A total of 1329 proteins were detected, with the vast majority shared between groups (1302; 98%), emphasizing the homogeneity of the samples.

VIBRA-CELL) for 30 s with pulses of 1 s on and 1 s off at an amplitude of 50%.

Protein concentration was measured with nanodrop at 280 nm, with concentrations ranging from 1.7 to 7.1 mg/mL. A total of 20 μg protein from each sample was digested with Lysyl Endopeptidase (Wako, Osaka, Japan) in a ratio of 1:100 w/w and incubated for 3 h at 37 °C. Samples were diluted 5× with digestion buffer (10% acetonitrile, 50 mM HEPES pH 8.5) and digested overnight at 37 °C with Trypsin (modified sequencing grade; Sigma) in a 1:50 w/w ratio. Digestion was quenched by the addition of 100 μL 2% trifluoroacetic acid (TFA).

The resulting soluble peptides in the supernatant were desalted and concentrated using the SOLAμ HRP plate (Thermo Fisher Scientific) with activation by 100% methanol, MS Buffer B (80% acetonitrile, 0.1% formic acid), followed by equilibration with 3% acetonitrile, 1% TFA. The sample was applied to the plate, followed by 2× wash with MS Buffer A (0.1% formic acid in water). Samples were eluted using 2 × 40% ACN, 0.1% formic acid. Between each liquid application, the plate was centrifuged at 1500 rpm for 1 min.

Eluted peptides were dried in a SpeedVac (Eppendorf concentrator plus) for 1 h at 60 °C. Dried peptides were resuspended in 12 μL Buffer A* (2% ACN, 1% TFA) containing iRT peptides (Biognosys) and peptide concentration was measured with nanodrop at 280 nm.

**Mass spectrometry analysis**. In all, 1 μg of peptide mixture from each sample was analyzed by online nanoscale liquid chromatography-tandem mass spectrometry (LC-MS/MS) in turn. Peptides were separated on a 15 cm C18-column (Thermo EasySpray ES804A) using an EASY-nLC 1200 system (Thermo Scientific). The column temperature was maintained at 30 °C.

The flow rate of the gradient was kept at 250 nl/min, and started at 6% MS buffer B, going to 23% MS buffer B in 68 min. This was followed by a 17 min step going to 38% MS buffer B, increasing to 60% MS Buffer B in 5 min and finally ramping up to 95% MS buffer B in 3 min, holding it for 7 min to wash the column.

The Q Exactive HF-X instrument (Thermo Scientific, Bremen, Germany) was run in data-dependent acquisition mode using a top 20 Higher-energy Collisional Dissociation (HCD)-MS/MS

method with the following settings. Scan range was limited to 350–1850 m/z. Full-scan resolution was set to 60,000 m/z, with an AGC target of 3e6 and a maximum injection time (IT) value of 50 ms. Peptides were fragmented with a normalized collision energy of 28, having a dynamic exclusion of 60 s, excluding unassigned ions and those with a charge state of 1, 6–8. MS/MS resolution was set at 15,000 m/z, with an AGC target of 1e5 and a maximum IT of 30 ms.

**Statistics and reproducibility**. All 34 raw LC-MS/MS data files were processed together using Proteome Discoverer version 2.4 (Thermo) with the use of Label-free quantitation in both the processing and consensus steps. In the processing step, Oxidation (M) and protein N-termini acetylation and met-loss were set as dynamic modifications, with cysteine carbamidomethyl set as static modification. All results were filtered with percolator, using a 1% false discovery rate, and Minora Feature Detector was used for quantitation. Two separate searches were run using these same workflows and SequestHT as database, one matching spectra against the reviewed human database from Uniprot (downloaded July 2020), and one matching against the Human Oral Microbiome Database http://www.homd.org/ (downloaded July 2020). Output from Proteome Discoverer version 2.4 was analyzed with Perseus version 1.6.5.0[31]. For quality assurance proteins identified with less than two peptides were filtered out. Only the proteins identified in all samples were used in Principal component analysis (PCA) and volcano plots. Volcano plots were made using a double-sided t-test. For hierarchical clustering, Euclidean distances and average linkage were used.

Analysis of enriched biological terms was performed in the Database for Annotation, Visualization and Integrated Discovery (DAVID) version 6.8. Significant differentially expressed proteins were mapped to Uniprot Accession. Three entries could not be mapped (P0DOX2, P0DOX5, and P0DOX7). Functional annotation was based on Kyoto Encyclopedia of Genes and Genomes (KEGG) pathway, Uniprot Keywords, and three subcategories of Gene Ontology: Molecular Function (GO:MF), Cellular Component (GO:CC), and Biological Process (GO:BP). Terms with Benjamini-adjusted P values < 0.05 were included.

**Metaproteomics analysis**. The relative intensity abundance of genera was calculated as the fraction of the summed intensities on genus level for the individual samples, compared to the total intensity. Only proteins with two or more peptides were included, and genera represented with less than one protein were excluded.

Functional annotation analysis of the bacterial proteins was done by mapping SEQID's from the human oral microbiome database to uniprot ID by similarity using PROKKA[32]. GO:BP terms were assigned using uniprot ID's and the intensity was summed for matching terms and the median for each group was used for calculating fractions of total intensity. Terms representing more than 0.2% abundances were manually merged to 11 terms.

**Reporting summary**. Further information on research design is available in the Nature Portfolio Reporting Summary linked to this article.

**Results**

**Clinical outcomes of AT-MSC therapy**. The clinical outcomes have been published[25]. No serious treatment-related adverse events occurred during the 120-day follow-up period. Treatment with AT-MSCs induced a significant increase in saliva flow rate. Specifically, the unstimulated whole-saliva flow rate increased from 0.13 ± 0.02 mL/min at baseline to 0.18 ± 0.02 mL/min at day

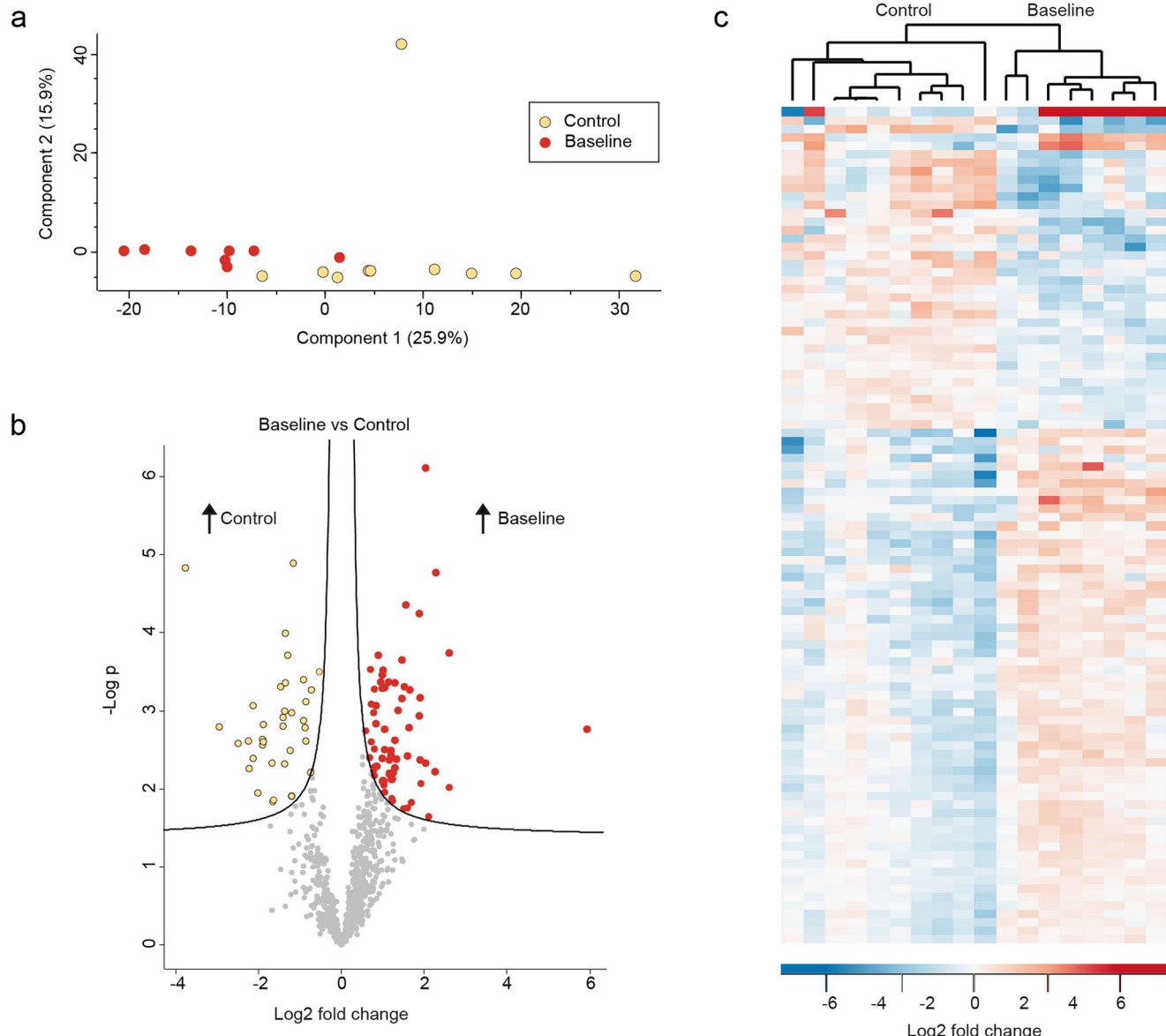

**Fig. 3 Comparison of healthy controls and baseline group. a** Principal component analysis (PCA) plot demonstrating clear separation in component 1. **b** Volcano plot presenting 140 significantly differentially expressed proteins. **c** Heatmap of the significantly differentially expressed proteins showing clustering of the two groups.

120 ([95% CI: 0.03 to 0.09]; $P = 0.0009$), whereas stimulated whole-saliva flow rate increased from $0.66 \pm 0.11$ mL/min at baseline to $0.75 \pm 0.11$ min at day 120 ([95%CI: 0.02 to 0.16]; $P = 0.017$). Patient-reported xerostomia (dry mouth symptoms) decreased by 22.5 units ($P = 0.0004$) measured by the Xerostomia Questionnaire (XQ) summary score[33].

**Protein expression**. A total of 34 samples of stimulated whole saliva were collected from 8 patients with previous oropharyngeal squamous cell carcinoma and with radiation-induced salivary gland hypofunction (three samples from each patient at baseline and on Day 5 and 120, 24 samples in all) and from 10 age-, sex- and educationally matched healthy controls. From the total of 34 samples, 1329 different proteins were identified, with a mean number of 1206 proteins (range: 1017–1276) per sample. As seen in Fig. 2, a total of 98% of the same proteins were identified in both the samples from the healthy controls and in patients treated with radiation therapy at baseline, day 5 and day 120. No

significant differences in the mean number of proteins were observed between the healthy controls (mean: 1171, range:1017–1260) and the radiation therapy group at baseline (mean: 1227, range 1181–1250), day 5 (mean: 1210, range: 1181–1249), or day 120 (mean: 1225, range:1171–1276) ($P > 0.05$).

**The salivary proteome of patients treated with radiation therapy differs from that of healthy controls**. PCA showed a clear separation of samples from patients treated with radiation therapy at baseline before the intervention versus healthy controls based on the most decisive variable (component 1) accountable for 25.9% of the variation (Fig. 3a). The volcano plot in Fig. 3b demonstrates the distribution of proteins in the two groups. In total, 140 proteins were significantly differentially expressed in patients treated with radiation therapy compared with the healthy controls. Specifically, 93 proteins were upregulated and 47 were downregulated in baseline samples from the patients as compared

**Table 1 Baseline salivary proteome from patients treated with radiotherapy compared to healthy controls.**

| Accession | Proteins with significantly higher intensity in healthy controls vs. baseline test group | −LOG (*P* value) | Difference |
|---|---|---|---|
| P31025 | Lipocalin-1 | 4.83 | −3.78 |
| P35754 | Glutaredoxin-1 | 2.80 | −2.96 |
| P01036 | Cystatin-S | 2.58 | −2.51 |
| P15515 | Histatin-1 | 2.61 | −2.25 |
| P09228 | Cystatin-SA | 2.26 | −2.22 |
| P02808 | Statherin | 2.39 | −2.13 |
| P28325 | Cystatin-D | 3.07 | −2.13 |
| P01037 | Cystatin-SN | 1.95 | −2.03 |

| Accession | Proteins with significantly lower intensity in healthy controls vs. baseline test group | −LOG (*P* value) | Difference |
|---|---|---|---|
| Q96JY6 | PDZ and LIM domain protein 2 | 2.33 | 2.03 |
| P52597 | Heterogeneous nuclear ribonucleoprotein F | 6.11 | 2.03 |
| O43399 | Tumor protein D54 | 1.65 | 2.11 |
| Q99497 | Protein/nucleic acid deglycase DJ-1 | 2.22 | 2.26 |
| P59666 | Neutrophil defensin 3 | 4.77 | 2.28 |
| P56385 | ATP synthase subunit e, mitochondrial | 3.74 | 2.60 |
| P62899 | 60 S ribosomal protein L31 | 2.02 | 2.60 |
| P84098 | 60 S ribosomal protein L19 | 2.76 | 5.92 |

**Table 2 Baseline salivary proteome from patients treated with radiotherapy compared to salivary proteome 120 days after AT-MSC therapy.**

| Accession | Proteins with significantly lower intensity in test griuo day 120 vs. test group at baseline | −LOG (*P* value) | Difference |
|---|---|---|---|
| P84098 | 60 S ribosomal protein L19 | 1.883 | 4.170 |
| Q9NR46 | Endophilin-B2 | 5.494 | 2.845 |
| P02808 | Statherin | 1.807 | 2.463 |
| Q9Y6R7 | IgGFc-binding protein | 1.935 | 2.018 |

| Accession | Proteins with significantly higher intensity in test group day 120 vs. test group at baseline | −LOG (*P* value) | Difference |
|---|---|---|---|
| P26583 | High mobility group protein B2 | 3.073 | −4.898 |
| Q86YZ3 | Hornerin | 2.553 | −3.940 |
| Q07157 | Tight junction protein ZO-1 | 2.967 | −3.496 |
| Q14019 | Coactosin-like protein | 3.061 | −3.315 |
| P02749 | Beta-2-glycoprotein 1 | 1.727 | −2.955 |
| P06870 | Kallikrein-1 | 2.592 | −2.953 |
| O60814 | Histone H2B type 1-K | 3.276 | −2.631 |
| Q13283 | Ras GTPase-activating protein-binding protein 1 | 4.404 | −2.585 |
| P35998 | 26 S proteasome regulatory subunit 7 | 7.769 | −2.535 |
| P02652 | Apolipoprotein A-II | 2.713 | −2.416 |
| P06737 | Glycogen phosphorylase | 2.481 | −2.342 |
| Q5D862 | Filaggrin-2 | 1.745 | −2.280 |
| P14317 | Hematopoietic lineage cell-specific protein | 2.566 | −2.204 |
| O00515 | Ladinin-1 | 2.729 | −2.189 |
| P61088 | Ubiquitin-conjugating enzyme E2 N | 4.852 | −2.153 |

to in healthy controls (Supplementary Data 1). Based on these proteins, clear separation of samples from each group was observed by means of cluster analysis (Fig. 3c). Sixteen proteins were identified with a twofold (log2) difference; eight were upregulated, and eight were downregulated in patients treated with radiation therapy compared to healthy controls (Table 1). In healthy controls, lipocalin-1 was the protein found with the highest difference (fold change) as compared to patients treated with radiation therapy. Also, salivary levels of four different cystatins (S, D, SA, and SN) were observed with higher intensities in healthy controls. On the other hand, 60 S ribosomal proteins L19 and L31 were the two proteins with the highest difference in intensity in saliva from patients treated with radiation therapy versus healthy controls. Analysis of functional annotation revealed enrichment of Salivary Secretion (KEGG pathway) in healthy controls but limited additional information (Supplementary Information File).

**AT-MSC therapy impacts the salivary proteome at day 120 but not at day 5.** Data on the human salivary proteome obtained from day 5 did not display any significant differences as compared to baseline samples (Supplementary Information File) and was not included in further analysis of human proteins. Based on the most decisive component (component 1: 22.6%) of the PCA, clear separation was observed between samples collected before treatment (baseline) and samples collected 120 days post treatment with AT-MSCs (Fig. 4a). A total of 36 proteins were identified with significantly higher intensities at baseline, whereas 63 proteins were seen to have significantly lower intensities in saliva at baseline as compared to 120 days post treatment (Fig. 4b and Supplementary Data 2). Cluster analysis showed almost separate clustering of samples from the two groups (Fig. 4c). After 120 days, 15 proteins were upregulated at least twofold (log2) in saliva as compared to baseline levels. On the contrary, four

proteins were recorded with a twofold (log2) reduction/down-regulation (60 S ribosomal protein L19, endophilin-B2, statherin, and IgGFc-binding protein) (Table 2). No apparent associations of functional annotations could be discerned (Supplementary Information File).

**AT-MSC therapy does not reverse radiation-induced changes of the salivary proteome.** PCA (Fig. 5a) showed clear separation between samples collected 120 days after AT-MSC therapy and samples from healthy controls, based on the most decisive component (component 1: 30.5%). A total of 212 proteins were identified with significantly different intensities, with 54 proteins associated with healthy controls and 158 proteins associated with samples collected at day 120 (Fig. 5b and Supplementary Data 3). In addition, cluster analysis showed clear separation of samples from healthy controls and day 120 samples from the AT-MSC-treated group (Fig. 5c). Eighteen proteins were upregulated at least a twofold (log2) in patients 120 days after AT-MSC therapy compared to healthy controls, while 11 were downregulated (Table 3). Seven proteins (statherin, lipocalin-1, glutaredoxin-1, histatin-1, cystatin-D, cystatin-S, and cystatin-SA) were identified with more than a twofold (log2) difference in healthy controls, when compared to patients treated with radiation therapy, both at baseline and 120 days post AT-MSC therapy. Likewise, hornerin, high mobility group protein B2, tight junction protein ZO-1, and ladinin-1 were all found with more than a twofold (log2) higher intensity in samples from patients 120 days after AT-MSC therapy, as compared to baseline samples in the patients and healthy controls, respectively. As observed at baseline, functional annotation analysis displayed enrichment of Salivary Secretion (KEGG pathway) in healthy controls compared to treated patients (Supplementary Information File).

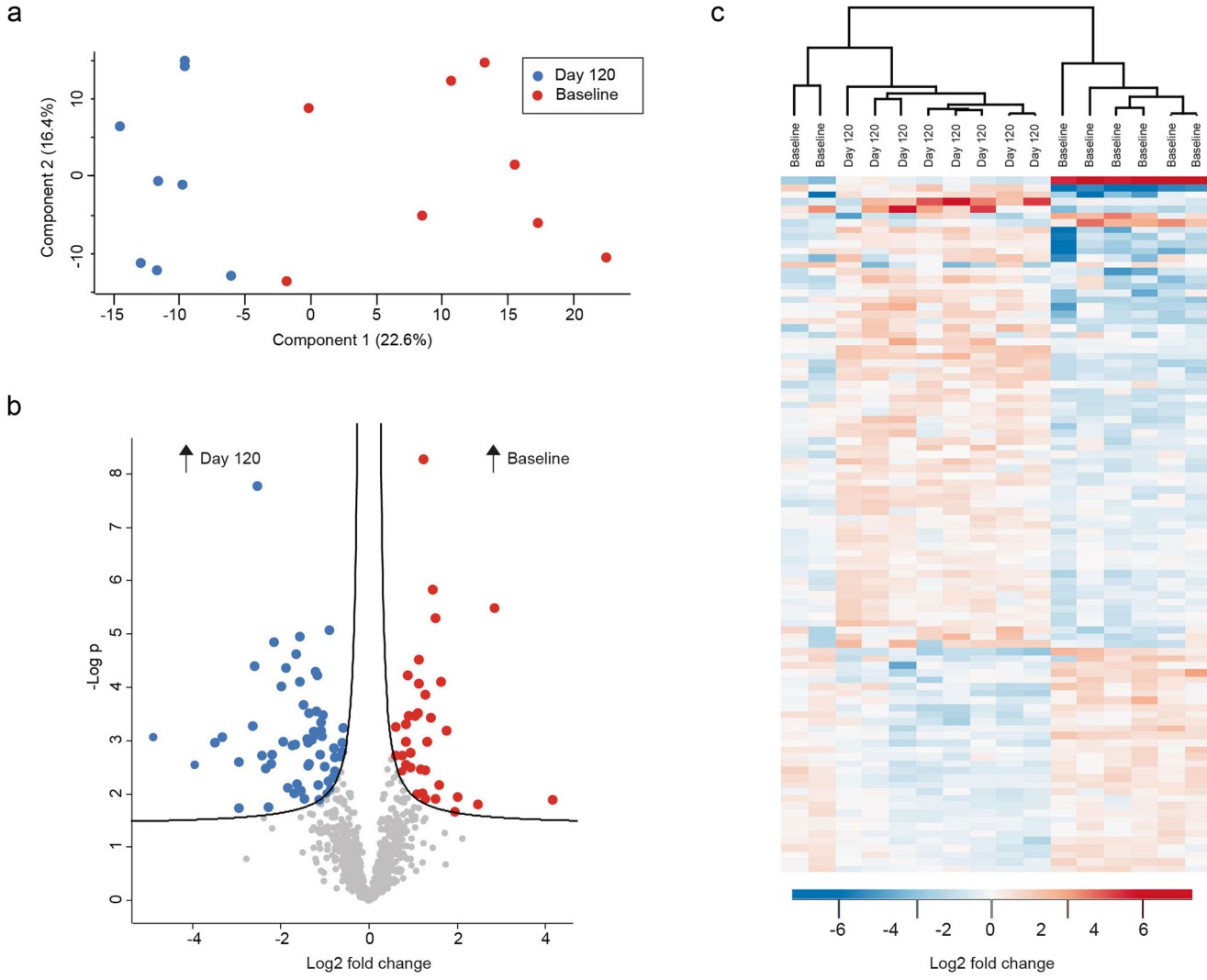

**Fig. 4 Comparison of day 120 and baseline group. a** Principal component analysis (PCA) plot showing clear separation in component 1. **b** Volcano plot visualizing 99 significantly differentially expressed proteins. **c** Heatmap of the significantly differentially expressed proteins demonstrating clustering of the two groups.

**The salivary microbiota is not influenced by AT-MSC therapy.** Figure 6a displays the relative abundance of the ten predominant bacterial genera identified by metaproteomics in saliva from patients treated with radiation therapy and healthy controls. In general, major interindividual variation was observed in both the radiation therapy group and the healthy control group. However, a higher abundance of Streptococcus species at the expense of an abundance of Neisseria species was seen in saliva from patients treated with radiation therapy, as compared with healthy controls, which was not affected by the AT-MSC therapy (Fig. 6a). In addition, no significant impact of AT-MSC therapy of functional expression of the oral microbiota was observed (Fig. 6b).

## Discussion

To the best of our knowledge, this trial is the first to use mass spectrometry to evaluate the mode of action of AT-MSCs in patients suffering from radiation-induced salivary gland hypofunction by examining the salivary proteome. The main finding was that treatment with locally injected allogeneic AT-MSCs induced marked changes to the salivary proteome, and that long-term effects could be demonstrated after 120 days. Among the differentially expressed proteins, several possess pleiotropic effects and could be involved in the regulation of cell growth and

development, immune system functions, angiogenesis, or regeneration. In addition, we report that radiation leads to long-term reduced production of the major salivary protein families, which provides new insights into the permanent changes occurring in the composition and quality of saliva after irradiation.

Through clinical trials, our group has previously demonstrated that autologous and allogeneic AT-MSCs injected into the salivary glands can enhance the unstimulated salivary flow rate in patients treated for oropharyngeal squamous cell carcinoma[18,19,25]. In this study, we report that treatment with allogeneic AT-MSCs administered directly into the parotid and submandibular glands leads to a significant increase in the intensity of 63 proteins in stimulated whole saliva 120 days after the intervention (Fig. 4b, c). Furthermore, AT-MSC therapy resulted in clinical improvement, as expressed by a significant increase in saliva secretion[25]. However, the proteins identified with increased intensity at day 120, compared to baseline, were not associated with classic functions in saliva. Instead, we identified upregulation of proteins involved in multiple intracellular and extracellular functions, such as tissue regeneration, including four proteins, high mobility group protein B2 (HMGB2), hornerin, tight junction protein ZO-1 (ZO-1) and coactosin-like protein, which increased more than threefold (log2) in intensity 120 days after AT-MSC administration (Table 2).

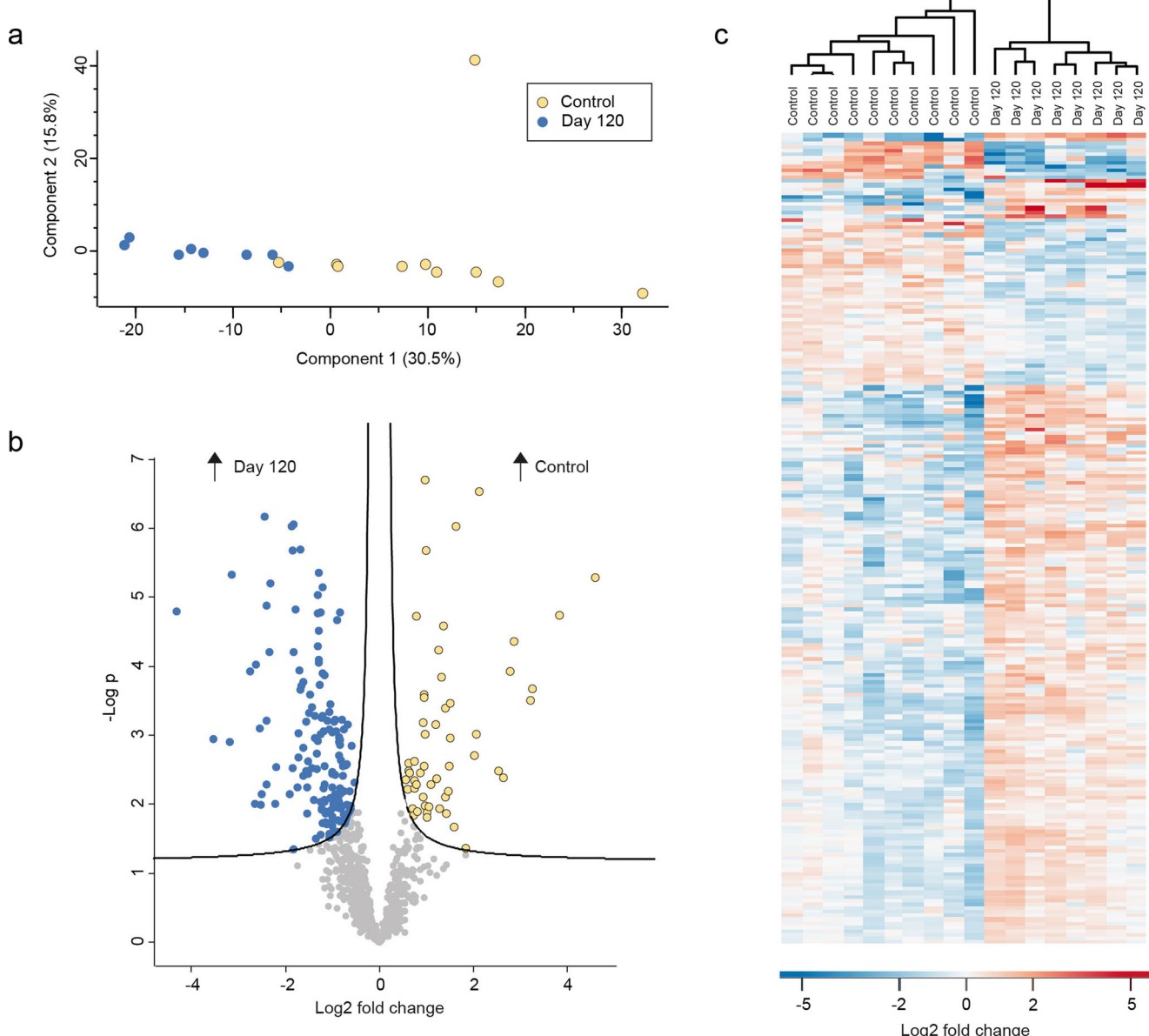

**Fig. 5 Comparison of healthy controls and day 120. a** Principal component analysis (PCA) plot demonstrating clear separation in component 1. **b** Volcano plot showing 212 significantly differentially expressed proteins. **c** Heatmap of the significantly differential expressed proteins validating clustering of the two groups.

While the presence of these distinct proteins is ambiguous in nature, data on all four indicate their relation to regeneration. HMGB2 is a DNA-binding protein which has been demonstrated to induce secretion of multiple cytokines associated with the recruitment of MSCs and with the regulation of cell development, growth and migration as well as stress response and immune system functions[34]. ZO-1 is, among others, expressed in all epithelial and endothelial cells and plays a vital role in the homeostasis of the endothelial cells by regulating cell–cell junction permeability and regenerating blood vessels. Moreover, it was demonstrated in vitro that MSCs could modulate the assembly of epithelial tight junctions in a process involving relocation of ZO-1[35]. Hornerin is a member of a group of proteins involved in the maintenance of calcium homeostasis, as well as other fundamental cellular processes and signaling cascades, and data support a relation between hornerin and wound regeneration[36,37]. Coactosin-like proteins are involved in cell growth and cell structure organization and have been detected in MSCs derived from amniotic fluid[38]. Interestingly,

animal studies indicate that MSCs including AT-MSCs have the potential to restore the function of salivary glands damaged by radiation therapy, as evidenced by increased salivary flow rate and indications of reduced fibrosis and improved blood flow[11,39]. Therefore, our data, which demonstrate an upregulation of the above-mentioned proteins in the salivary proteome of patients treated with AT-MSCs, may infer that AT-MSCs induce a wide range of processes to support glandular homeostasis and regeneration, indicating an ongoing regenerative process in the salivary glands. However, it is important to stress that these proteins are involved in multiple biological processes, which is why further studies are called for to investigate whether the upregulation of these proteins is in fact proof of tissue regeneration.

Saliva is an attractive medium for the identification of candidate biomarkers, because it can be collected easily and noninvasively. Therefore, saliva can be used to monitor clinical interventions such as AT-MSC therapy[40,41]. In line, the salivary proteome has been suggested as a resource of potential biomarkers of oral cancer, and

**Table 3 Salivary proteome 120 days after AT-MSC therapy compared to healthy controls.**

| Accession | Proteins with significantly higher intensity in healthy controls vs. day 120 test group | −LOG (P value) | Difference |
|---|---|---|---|
| P02808 | Statherin | 5.284 | 4.597 |
| P31025 | Lipocalin-1 | 4.737 | 3.831 |
| P35754 | Glutaredoxin-1 | 3.669 | 3.255 |
| P15515 | Histatin-1 | 3.500 | 3.214 |
| P02814 | Submaxillary gland androgen-regulated protein 3B | 4.365 | 2.871 |
| P28325 | Cystatin-D | 3.919 | 2.781 |
| P01036 | Cystatin-S | 2.380 | 2.645 |
| P09228 | Cystatin-SA | 2.479 | 2.539 |
| Q71DI3 | Histone H3.2 | 6.532 | 2.118 |
| P07711 | Cathepsin L1 | 3.018 | 2.070 |
| Q5JTV8 | Torsin-1A-interacting protein 1 | 2.705 | 2.024 |

| Accession | Proteins with significantly lower intensity in healthy controls vs. day 120 test group | −LOG (P value) | Difference |
|---|---|---|---|
| Q86YZ3 | Hornerin | 4.795 | −4.322 |
| P26583 | High mobility group protein B2 | 2.942 | −3.541 |
| Q07157 | Tight junction protein ZO-1 | 2.898 | −3.189 |
| O00515 | Ladinin-1 | 5.322 | −3.148 |
| P61313 | 60 S ribosomal protein L15 | 3.932 | −2.748 |
| P20810-6 | Isoform 6 of Calpastatin | 2.000 | −2.639 |
| P56385 | ATP synthase subunit e, mitochondrial 2 | 4.020 | −2.635 |
| Q99497 | Protein/nucleic acid deglycase DJ-1 | 3.101 | −2.555 |
| P62899 | 60 S ribosomal protein L31 | 1.986 | −2.530 |
| P03973 | Antileukoproteinase | 2.150 | −2.511 |
| P31946-2 | Isoform Short of 14-3-3 protein beta/alpha | 6.162 | −2.445 |
| Q9BUF5 | Tubulin beta-6 chain | 2.285 | −2.407 |
| P59666 | Neutrophil defensin 3 | 4.878 | −2.406 |
| P50991 | T-complex protein 1 subunit delta | 3.206 | −2.401 |
| P49207 | 60 S ribosomal protein L34 | 4.200 | −2.349 |
| P49913 | Cathelicidin antimicrobial peptide | 5.206 | −2.326 |
| Q04837 | Single-stranded DNA-binding protein, mitochondrial | 2.006 | −2.226 |
| P09497-2 | Isoform Non-brain of Clathrin light chain B | 2.534 | −2.195 |

the human salivary proteome is reported to differentiate patients with periodontitis and Sjögren's syndrome from healthy controls[42–44]. Indeed, much attention has been paid to the human salivary proteome, and in 2021, the community-driven research platform "The Human Salivary Proteome Wiki" was launched[45]. Importantly, the whole salivary proteome is a conglomerate of proteins of various origins, with the parotid, submandibular and sublingual major salivary glands as well as the minor salivary glands, and the gingival crevicular fluid, blood plasma, and oral microbiota as main donors[45]. Thus, alterations of the protein content in saliva will in many cases not portray alterations of concomitant proteins in salivary gland tissue, in detail, and therefore data on the whole-saliva proteome should always be considered exploratory, and findings should preferably be validated in the tissue of interest. We used stimulated whole-saliva samples and did not attempt to collect sterile samples separately from the parotid (e.g., by Lashley cup) or submandibular glands, as because these samples are often contaminated, and because only whole saliva offers information about the oral microbiota. In addition, it is our experience that the collection of both whole-saliva samples, and samples from the separate salivary glands is too extensive for the majority of irradiated patients with the risk of burdening the

patients and having samples with questionable value[18]. Furthermore, obtaining tissue biopsies from the glands was not feasible due to the limited amount of remaining salivary gland tissue in irradiated patients. Therefore, whole-saliva sampling with its known limitations was considered the best proxy for evaluating the impact of AT-MSC therapy on the salivary gland function.

Interestingly, our data show that the composition of the human proteome of whole saliva in patients with radiation-induced salivary gland damage differs significantly from that of healthy matched controls years after radiation therapy. Accordingly, 140 proteins were significantly different between control persons and patients with a median of 5.5 years after radiation therapy (Fig. 3a). Six of the ten proteins with more than twofold (log2) higher intensity in the healthy control persons were from major salivary protein families with specific activities in normal saliva: four cystatins, statherin, and histatin-1 (Table 1). Cystatins and histatins act as an important first-line of defense against microorganisms in the oral cavity[46,47]. Lower levels of cystatin-S in saliva have been reported in children with caries as well as in patients with Sjögren's syndrome versus healthy controls, whereas salivary cystatin-SA levels associate with periodontal status[48,49]. In addition, salivary levels of cystatin-D affects the subjective perception of oral dryness, and different parotid levels of cystatins and histatins has previously been reported in two irradiated patients versus two healthy controls[50,51]. Thus, our data are in line with previous reports underlining the important role of salivary Cystatins in oral health and disease. Previous data indicate an important role of statherin in calculus formation, and salivary levels of statherin associates with oral hygiene status[52,53]. Histatin-1 has predominately anticandidal activity, and it is well known that patients with previous head and neck cancer often suffer from oral candida infections as a side effect of radiation therapy, which is in line with a very recent paper reporting substantial changes in the salivary proteome after radiotherapy of head and neck cancer patients, including a decrease in salivary levels of cystatins and histatins[3,54]. Consequently, our data underline that radiation not only has a detrimental impact on saliva secretion in terms of volume but also affects the production of specific proteins, which all are critically involved in the maintenance of oral homeostasis conveyed by saliva.

Although AT-MSC therapy induced significant changes in the human saliva proteome in the patients at day 120, the proteome remained significantly different from that of matched healthy controls (Fig. 5a–c). In general, upregulation and downregulation of the same proteins were identified in baseline and day 120 samples in the radiation group, as compared to the healthy controls. Therefore, our data clearly demonstrate that AT-MSC therapy does not reverse the structural alterations of the salivary proteome caused by radiation. Several factors may have influenced this finding. First, the changes were analyzed just 4 months after the intervention. Thus, it is possible that potential tissue remodeling induced by the AT-MSCs was still an ongoing process, which could, in time, further improve the composition of the salivary proteome. To test this, we therefore plan to repeat sample collection 1 and 3 years after the intervention for further proteomic analyses. Second, we treated patients who had received radiation therapy at least 2 years before the AT-MSC therapy. Theoretically, there could be more biological potential in the treatment of patients closer to their radiation therapy at a time point involving less fibrosis and more active inflammation prone to treatment with AT-MSCs.

In this study, we identified significant differences in the composition of the salivary microbiota of irradiated patients as compared to healthy controls. On the other hand, treatment with AT-MSCs had no impact on the composition of the salivary microbiota, as evaluated 120 days after the intervention (Fig. 6a).

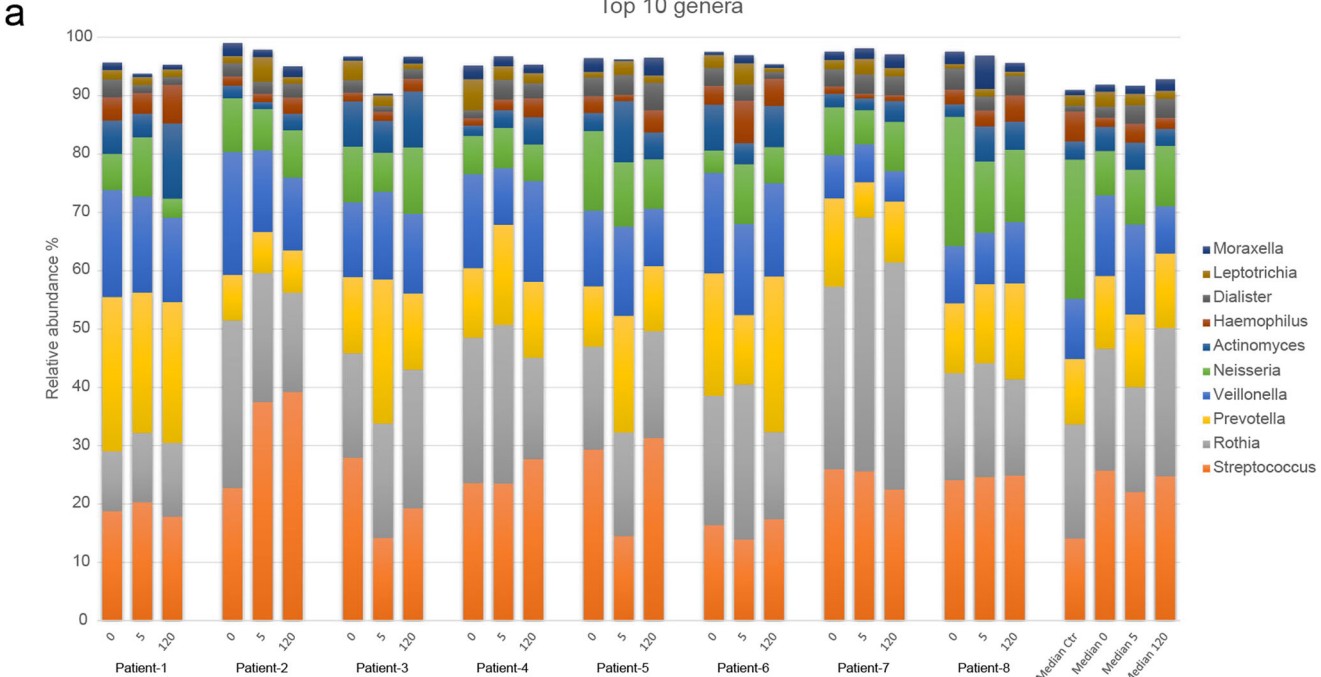

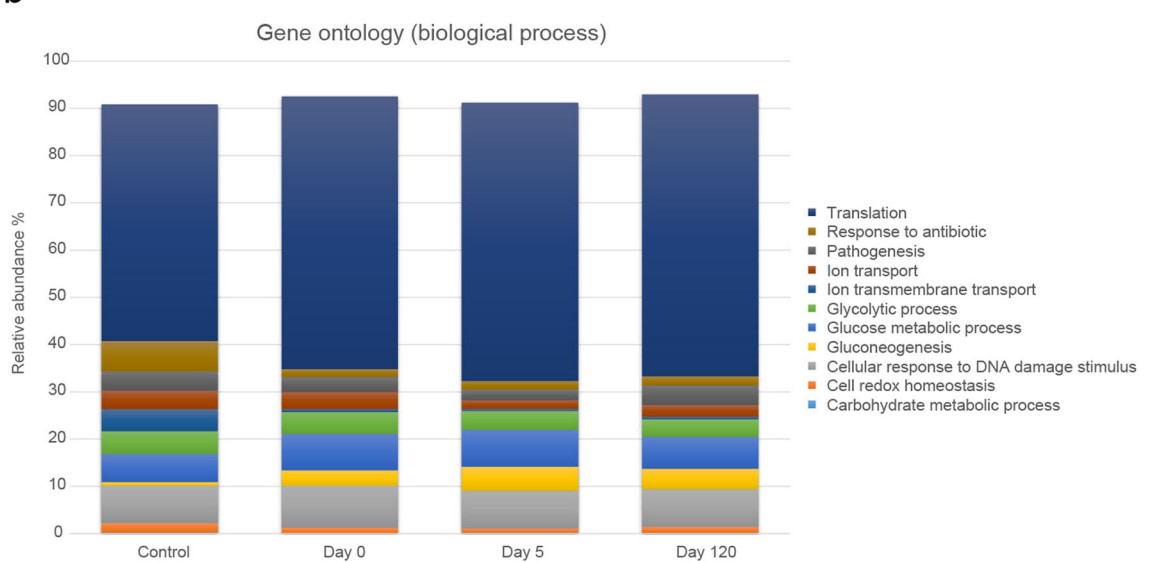

**Fig. 6 Metaproteomics data. a** Top ten genera expressed in the patient's timeline and the median in the groups. The relative abundance is shown as the fraction of the summed intensities. **b** Functional annotation of bacterial proteins in the four groups.

Likewise, AT-MSC treatment had no effect on the functional expression of the salivary microbiota (Fig. 6b). Therefore, when considering the impact AT-MSC on the composition of the salivary proteome 120 after treatment (Fig. 4), our data suggest that the composition of the salivary microbiota is shaped predominantly by the volume of saliva being secreted, rather than the concentration of specific proteins in saliva. Specifically, we report higher levels of Streptococcus species and decreased levels of *Neisseria* species in the irradiated patients, as compared to the controls, which is in line with previous findings[55]. Importantly, due to their proficient carbohydrate metabolism, Streptococcus species is central players in the pathogenesis of caries[56,57]. Consequently, our findings of increased salivary levels of Streptococcus species in saliva from irradiated patients, which is in line with previous findings, might be one part of the biological

puzzle, which in combination with impairment of the natural salivary defense systems, can explain the increased frequency of dental caries observed in patients treated with radiation[56,58,59].

The main limitation of this exploratory study is the small sample size. Nevertheless, we were able to identify significant alterations to the salivary proteome 120 days after AT-MSC treatment, which could reflect lasting changes induced by the AT-MSCs in the salivary glands. A second limitation is the lack of a placebo-controlled comparison. However, to address this issue, our mass spectrometry analysis was performed in a blinded fashion. Finally, we did not centrifuge the samples prior to analysis, which entails that the cells were not removed from the samples. We chose this strategy to enable concomitant characterization of both the human and the bacterial proteome. However, the human proteins identified may originate from a

wide variety of cells, including desquamated epithelial cells. Given the fact that the AT-MSC therapy did not seem to impact the salivary microbiota, a centrifugation step could be included in further studies to reduce variation, e.g., from differences in the number of cells present in samples, and ease the interpretation of the human proteome by excluding strictly intracellular proteins. Naturally, our findings should be validated in larger studies with longer follow-up periods.

In conclusion, the current study provides important insights on the mode of action that allogeneic AT-MSCs induce in radiation-damaged salivary glands, as treatment with allogeneic AT-MSCs in irradiated patients promotes an increase in proteins related to tissue regeneration in stimulated whole saliva. This study also provides new insights into how irradiation manifests in the proteome of whole saliva years after radiation treatment.

## Data availability

The mass spectrometry proteomics data have been deposited to the ProteomeXchange Consortium via the PRIDE partner repository with the dataset identifier PXD024152[60] (https://www.ebi.ac.uk/pride/archive). All other data supporting the findings of this study are available from the corresponding authors upon reasonable request. Source data for the figures are available as Supplementary Data 4.

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

## Acknowledgements
We thank the patients and healthy controls who participated in this study. We would also like to thank colleagues who provided assistance or instruction Anne Kathrine Østergaard Madsen, Camilla Bucken Campen, Mandana Haack-Sørensen. Charlotte Duch Lynggaard would like to acknowledge the support from the non-profit Candys Foundation (grant 2017-218).

## Author contributions
All authors agree to the submission of this manuscript and are responsible for the integrity and ethics of the methods or data they contributed. C.D.L., R.J.C., C.G., S.B.J., S.J., M.H., A.F.N., J.M., M.M.H., A.E., J.K, D.B., and C.V.B. contributed to the experimental design. R.C.J. and M.J. prepared samples for sequencing, analyzed and interpreted the data with input from C.D.L., S.B.J., C.V.B., and D.B. C.D.L., R.J.C., M.J., C.V.B., and D.B. wrote the manuscript, which was approved by all co-authors.

## Competing interests
The authors C.D.L., C.G., and C.V.B. declare the following competing interests: Rigshospitalet and the University of Copenhagen have a pending patent application entitled "Stem cell therapy for salivary gland dysfunction" published as WO 2020/165405 A1, with C.D.L., C.G., and C.V.B. as the inventors. The authors A.E. and J.K. declare the following competing interests: A.E. and J.K. are inventors of the patent "Stem cell therapy based on adipose-derived stem cells" (Publication WO 2017-068140). All authors met the ICMJE authorship criteria. No honoraria or payments were made for authorship. The remaining authors declare no competing interests.
