## [Peer Review File · Communications Medicine]

Reviewers' comments:

Reviewer #1 (Remarks to the Author):

The manuscript "Intraglandular mesenchymal stem cell treatment induces change in the salivary proteome of irradiated patients" by Lynggaard and colleagues describes the results of interesting experiments carried out with the aim to alleviate radiation-induced hyposalivation in patients suffering for head and neck cancer, using an intra-glandular inoculation of a preparation of adipose tissue-derived mesenchymal stem/stromal cells. The treatment effectiveness were evaluated analysing the salivary proteomes, the salivary flow rate and partly the microbiota of the patients before and after the treatment and comparing them with those of healthy controls. Results indicated that the treatment slightly improves the salivary secretion of several proteins. Nonetheless, the treatment is far to restore the salivary proteome of healthy adults, matched for age and gender.

Although the results were not completely satisfactory, they are interesting and the manuscript deserves publication. The experimental plan was carried out correctly, the proteomic platforms utilized, data analyses and figures were adequate. My advise is to improve the references relative to human salivary proteome, which have been poorly reported and discussed.

Reviewer #2 (Remarks to the Author):

Interesting, pilot study from a leader in work investigating novel treatments for radiation-induced xerostomia.

My main suggestion is to clarify in the discussion/methods whether and how you normalized for more concentrated saliva in irradiated patients compared to baseline and post-MSC treatment. If MSC results in more saliva production as shown does this suggest proteins will be more dilute even if there is no change in protein production?

Figure 2 – please add number of patients (n=X) for each sample condition (baseline, control etc...)

Figure 6A – it is unclear from the figure and legend whether each group of 3 bars represents an individual patient or something else – please clarify –(perhaps by labeling each group at the top (Pt-1, Pt2, etc...))

Please update reference for MESRIX_II study (reviewer recognizes this paper was submitted prior to publication date).

Reviewer #3 (Remarks to the Author):

Intraglandular mesenchymal stem cell treatment induces change in the salivary proteome of irradiated patients by Charlotte Duch Lynggaard et al.

The authors studied the proteome and microbiome of saliva in previously radiated head-and-neck cancer patients after treatment with autologous adipose tissue-derived mesenchymal stem/stromal cells (AT-MSCs) to the submandibular and parotid salivary glands. It was observed that AT-MSC treatment significantly affected protein expression 120 days post baseline. However, compared to healthy controls, it was documented that AT-MSC treatment did not restore the proteome to that of healthy conditions.

This is an interesting work and a well written manuscript that is easy to follow. As the authors themselves indicate, the study group is too small to make any specific conclusions. But, nonetheless, this is a promising start.

Since the manuscript is presented very clearly and the work appears to be of high quality, I only have a few comments:

1. Page 4: Last sentence of first paragraph. Seems misplaced.
2. Page 5: Please explain how you ended up with 34 samples. And in the same sentence, please specify that this is days after AT-MSC treatment.
3. Page 5, under "The salivary proteome of patients treated with radiation therapy differs from that of healthy controls": Consider including all significantly up- or downregulated proteins in a supplementary table. In fact, I did not find any included tables in the submitted material.
4. Page 9, comment on saliva sampling: How are the possibilities of collecting at least parotid saliva separately?
5. Page 13, under "Study population": Please include the radiation doses.
6. Figure 1: Please include in the legend additional explanation of the steps included in the figure. It is not entirely evident for all readers what the various illustrations refer to. And change "choreography".

Response to the editors and reviewers

First, we would like to thank the reviewers for providing very helpful comments and dedicating their time to improving our manuscript. Our answers are marked in blue and our changes in the manuscript are marked in red.

Reviewer 1

The manuscript “Intraglandular mesenchymal stem cell treatment induces change in the salivary proteome of irradiated patients” by Lynggaard and colleagues describes the results of interesting experiments carried out with the aim to alleviate radiation-induced hyposalivation in patients suffering for head and neck cancer, using an intra-glandular inoculation of a preparation of adipose tissue-derived mesenchymal stem/stromal cells. The treatment effectiveness was evaluated analysing the salivary proteomes, the salivary flow rate and partly the microbiota of the patients before and after the treatment and comparing them with those of healthy controls. Results indicated that the treatment slightly improves the salivary secretion of several proteins. Nonetheless, the treatment is far to restore the salivary proteome of healthy adults, matched for age and gender. Although the results were not completely satisfactory, they are interesting and the manuscript deserves publication. The experimental plan was carried out correctly, the proteomic platforms utilized, data analyses and figures were adequate. My advice is to improve the references relative to human salivary proteome, which have been poorly reported and discussed.

1. Q: My advice is to improve the references relative to human salivary proteome, which have been poorly reported and discussed.

Reply: Thank you for the appropriate advice and for the chance to elaborate on this.

Action: We have amended the Discussion as shown below and updated the references with the following:

Discussion (page 10+ 11)

Saliva is an attractive medium for the identification of candidate biomarkers, because it can be collected easily and non-invasively. Therefore, saliva can be used to monitor clinical interventions such as AT-MSc therapy^{32,33}. In line, the salivary proteome has been suggested as a resource of potential biomarkers of oral cancer, and the human salivary proteome is reported to differentiate patients with periodontitis and Sjögren’s syndrome from healthy controls^{34–36}. Indeed, much attention has been paid to the human salivary proteome, and in 2021, the community driven research platform “The Human Salivary Proteome Wiki” was launched³⁷. Importantly, the whole salivary proteome is a conglomerate of proteins of various origins, with the parotid, submandibular and sublingual major salivary glands, as well as the minor salivary glands and the gingival crevicular fluid, blood plasma and oral microbiota as main donors³⁷. Thus, alterations of the protein content in saliva will in many cases not portray alterations of concomitant proteins in salivary gland tissue, in detail, and therefore data on the whole saliva proteome should always be considered exploratory, and findings should preferably be validated in the tissue of interest.

We used stimulated whole saliva samples and did not attempt to collect sterile samples separately from the parotid (e.g., by Lashley Cup) or submandibular glands, as these samples are often contaminated, and because only whole saliva offers information about the oral microbiota. In addition, it is our experience that the collection of both whole saliva samples, and samples from the separate saliva glands is too extensive for the majority of irradiated patients with the risk of burdening the patients and having samples with questionable value³⁸.

Furthermore, obtaining tissue biopsies from the glands was not feasible due to the limited amount of remaining salivary gland tissue in irradiated patients. Therefore, whole saliva sampling with its known limitations was considered the best proxy for evaluating the impact of AT-MSCT therapy on the salivary gland function.

Discussion (page 11)

Lower levels of cystatin S in saliva have been reported in children with caries as well as in patients with Sjögren's syndrome versus healthy controls, whereas salivary cystatin SA levels associate with periodontal status^{41,42}. In addition, salivary levels of cystatin D affects the subjective perception of oral dryness, and different parotid levels of cystatins and histatins has previously been reported in two irradiated patients versus two healthy controls^{43,44}. Thus, our data are in line with previous reports underlining the important role of salivary Cystatins in oral health and disease. ~~Statherin is involved in mineralization, antibacterial defense, and lubrication of the mucosa.~~ Previous data indicate an important role of statherin in calculus formation, and salivary levels of statherin associates with oral hygiene status^{46,47}. Histatin-1 has predominately anticandidal activity, and it is well known that patients with previous head and neck cancer often suffer from oral candida infections as a side effect of radiation therapy, which is in line with a very recent paper reporting substantial changes in the salivary proteome after radiotherapy of head and neck cancer patients, including a decrease in salivary levels of cystatins and histatins^{48,49}. Consequently, our data ~~points out~~ underlines that radiation not only has detrimental impact on saliva secretion in terms of volume, but also effects the production of specific proteins, which all are critically involved in maintenance of oral homeostasis conveyed by saliva.

New references

Sivadasan, P. et al. Human salivary proteome — a resource of potential biomarkers for oral cancer. *J. Proteomics* 127, 89–95 (2015).

Sembler-Møller, M. L., Belstrøm, D., Loch, H. & Pedersen, A. M. L. Proteomics of saliva, plasma, and salivary gland tissue in Sjögren's syndrome and non-Sjögren patients identify novel biomarker candidates. *J. Proteomics* 225, 103877 (2020).

Belstrøm, D. et al. Metaproteomics of saliva identifies human protein markers specific for individuals with periodontitis and dental caries compared to orally healthy controls. *PeerJ* 4, e2433 (2016).

- Lau, W. W., Hardt, M., Zhang, Y. H., Freire, M. & Ruhl, S. The Human Salivary Proteome Wiki: A Community-Driven Research Platform. *J. Dent. Res.* **100**, 1510–1519 (2021).
- Martini, D. *et al.* Cystatin S-a candidate biomarker for severity of submandibular gland involvement in Sjögren's syndrome. *Rheumatology* **56**, 1031–1038 (2017).
- Techatanawat, S. *et al.* Salivary and serum cystatin SA levels in patients with type 2 diabetes mellitus or diabetic nephropathy. *Arch. Oral Biol.* **104**, 67–75 (2019).
- Yamamoto, K., Hiraishi, M., Haneoka, M., Fujinaka, H. & Yano, Y. Deep learning-based dental plaque detection on primary teeth: a comparison with clinical assessments. *BMC Oral Health* **20**, 1–7 (2020).
- Laheij, A. M. G. A. *et al.* Proteins and peptides in parotid saliva of irradiated patients compared to that of healthy controls using SELDI-TOF-MS. *BMC Res Notes* **8**, 639 (2015).
- Gowda, D. *et al.* Correlation of Salivary Statherin and Calcium Levels with Dental Calculus Formation: A Preliminary Study. *Int. J.* (2017).
- Gowda, D. *et al.* Association of Salivary Statherin, Calcium, and Proline-Rich Proteins on Oral Hygiene: A Cross-Sectional Study. *Int. J. den* (2021).
- Mendes, T. *et al.* Radiotherapy changes the salivary proteome in head and neck cancer patients: evaluation before, during, and after treatment. *Clin. Oral Investig.* **26**, 225–258 (2022).
- Pinna, R., Campus, G., Cumbo, E., Mura, I. & Milia, E. Xerostomia induced by radiotherapy: an overview of the physiopathology, clinical evidence, and management of the oral damage. *Ther. Clin. Risk Manag.* **11**, 171–88 (2015).

Reviewer 2

Interesting, pilot study from a leader in work investigating novel treatments for radiation-induced xerostomia.

My main suggestion is to clarify in the discussion/methods whether and how you normalized for more concentrated saliva in irradiated patients compared to baseline and post-MSCT treatment. If MSCT results in more saliva production as shown does this suggest proteins will be more dilute even if there is no change in protein production?

1. Q: My main suggestion is to clarify in the discussion/methods whether and how you normalized for more concentrated saliva in irradiated patients compared to baseline and post-MSCT treatment. If MSCT results in more saliva production as shown does this suggest proteins will be more dilute even if there is no change in protein production?

Reply: Thank you for suggesting this. The normalization mode in Proteome Discoverer were 'Total Peptide Amount' which sums the peptide group abundances for each sample and determines the maximum sum for all files. The normalization factor is the factor of the sum of the sample and the maximum sum in all files. After calculating the normalization factors, the Precursor Ions Quantifier node normalize consensus features by dividing abundances with the normalization factor over all samples.

In this way we feel confident that we did the best to take out any discrepancy between protein concentrations.

Action: We have updated the methods section with the sentence in red seen below

Data analysis

All 34 raw LC-MS/MS data files were processed together using Proteome Discoverer version 2.4 (Thermo) with the use of Label-free quantitation in both the processing and consensus steps. In the processing step, Oxidation (M) and protein N-termini acetylation and met-loss were set as dynamic modifications, with cysteine carbamidomethyl set as static modification. All results were filtered with percolator, using a 1% false discovery rate, and Minora Feature Detector was used for quantitation. **The abundances were globally normalized based on total peptide sum.** Two separate searches were run using these same workflows and SequestHT as database, one matching spectra against the reviewed human database from Uniprot (downloaded July 2020), and one matching against the Human Oral Microbiome Database <http://www.homd.org/> (downloaded July 2020). Output from Proteome Discoverer version 2.4 was analyzed with Perseus version 1.6.5.0 48 . For quality assurance proteins identified with less than two peptides were filtered out. Only the proteins identified in all samples were used in PCA and volcano plots. Volcano plots were made using a double-sided t-test. For hierarchical clustering, Euclidean distances and average linkage were used.

2. Q: Figure 2 – please add number of patients (n=X) for each sample condition (baseline, control etc...)

Reply: Thank for your relevant comment.

Action: We have updated the figure with the numbers of patients to the figure (see below).

Figure 2

3. Q: Figure 6A – it is unclear from the figure and legend whether each group of 3 bars represents an individual patient or something else – please clarify –(perhaps by labelling each group at the top (Pt-1, Pt2, etc...))

Reply: Thank for you for noticing this.

Action: We have added patient numbers to the bars (see below)

Figure 6

A)

4. Q: Please update reference for MESRIX_II study (reviewer recognizes this paper was submitted prior to publication date).

Reply: Thank you for pointing this out.

Action: We have updated the references with the published paper.

Lynggaard, C. D. *et al.* Intraglandular Off-the-Shelf Allogeneic Mesenchymal Stem Cell Treatment in Patients with Radiation-Induced Xerostomia : A Safety Study (MESRIX-II). *Stem Cells Transl. Med.* **11**(5), 478–489 (2022).

Reviewer 3

Intraglandular mesenchymal stem cell treatment induces change in the salivary proteome of irradiated patients by Charlotte Duch Lynggaard *et al.* The authors studied the proteome and microbiome of saliva in previously radiated head-and-neck cancer patients after treatment with autologous adipose tissue-derived mesenchymal stem/stromal cells (AT-MSCs) to the submandibular and parotid salivary glands. It was observed that AT-MSC treatment significantly affected protein expression 120 days post baseline. However, compared to healthy controls, it was documented that AT-MSC treatment did not restore the proteome to that of healthy conditions.

This is an interesting work and a well written manuscript that is easy to follow. As the authors themselves indicate, the study group is too small to make any specific conclusions. But, nonetheless, this is a promising start. Since the manuscript is presented very clearly and the work appears to be of high quality, I only have a few comments:

1. Q: Page 4: Last sentence of first paragraph. Seems misplaced.

Reply: Thank you for spotting this. We agree and have moved the sentence in the manuscript as seen below.

Action:

Recently, our group published findings from a randomized controlled trial showing promising results after treatment with autologous adipose tissue-derived mesenchymal stem/stromal cells (AT-MSCs) of the submandibular glands in patients treated for an oropharyngeal squamous cell carcinoma, including increased unstimulated salivary flow rate^{18,19}. **However, our first study did not elucidate the mechanism by which AT-MSCs enhanced the function of the salivary glands. Also, whereas our initial study focused on autologous AT-MSCs, the present study investigates treatment with allogeneic cells from healthy donors. As a benefit, the patients themselves avoid liposuction to obtain AT-MSCs for culture expansion, and the cell product is thoroughly characterized and standardized. However, our first study did not elucidate the mechanism by which AT-MSCs enhanced the function of the salivary glands.**

2. Q: Page 5: Please explain how you ended up with 34 samples. And in the same sentence, please specify that this is days after AT-MSC treatment.

Reply: Thank you for this comment. We have amended the sentence regarding the 34 samples as seen below. We hope the sentence is more understandably.

Action: A total of 34 samples of stimulated whole saliva were collected from 8 patients with previous oropharyngeal squamous cell carcinoma and with radiation-induced salivary gland hypofunction (3 samples from each patient at baseline and on Day 5 and 120, 24 samples in all) and from 10 age-, sex- and educationally-matched healthy controls.

3. Q: Page 5, under “The salivary proteome of patients treated with radiation therapy differs from that of healthy controls”: Consider including all significantly up- or downregulated proteins in a supplementary table. In fact, I did not find any included tables in the submitted material.

Reply: Thank you for noticing this. This is a mistake by us when submitting the manuscript initially. We have since uploaded tables and supplementary tables of all up- and downregulated proteins.

Action: Tables and supplementary tables are uploaded.

4. Q: Page 9, comment on saliva sampling: How are the possibilities of collecting at least parotid saliva separately?

Reply: Thank you for giving us the chance to elaborate on this. We agree that the collection of sterile saliva from the parotid and the submandibular glands could have been an option. For several reasons, we chose only to collect the whole saliva. We find whole saliva samples have several advantages over collecting saliva from the separate glands. In our experience collecting parotid saliva e.g. with a Lashley cup is complicated in patients with radiation-induced salivary gland hypofunction and even with this maneuver, the saliva is often contaminated. Furthermore, saliva from the parotid gland is sterile thereby excluding the opportunity to investigate the microbiome.

In our experience, it is too extensive for the majority of irradiated patients to participate in a collection of both whole saliva samples and separate saliva samples rendering the risk of burdening our patients and having saliva samples with questionable value.

Action: We have amended the discussion on page 9 with the following:

We used stimulated whole saliva samples and did not attempt to collect sterile samples separately from the parotid (e.g., by Lashley cup) and submandibular glands as these samples are often contaminated, and whole saliva offers information about the oral microbiome. In addition, it is our experience that collection of both whole saliva samples and samples from the separate salivary glands is too extensive for the majority of irradiated patients with the risk of burdening the patients and having samples with questionable value.

5. Q: Page 13, under “Study population”: Please include the radiation doses.

Reply: Thank you for this comment.

Action: We have amended the study population with the information below:

The patients had previously received photon therapy to a full dosage to the tumor and lymph node metastases of 66-68 Gray delivered in 2 Gray per fraction with 6 fractions per week concurrent with cisplatin therapy. This radiation therapy practically delivers the whole prescribed radiation dose to the ipsilateral submandibular gland and a minor fraction dose to the lower portion of the ipsilateral parotid gland.

6. Q: Figure 1: Please include in the legend additional explanation of the steps included in the figure. It is not entirely evident for all readers what the various illustrations refer to. And change “choreography”.

Reply: Thank you for your sharp observation and relevant suggesting.

Action: We have updated the figure with numbers 1 to 3 and amended the legend as seen below:

Fig. 1: Schematic depiction illustrating the analyses of stimulated whole saliva from patients treated with mesenchymal stem/stromal cells. Adipose tissue-derived mesenchymal stem/stromal cells (AT-MSCs) from three healthy female donors were used as a study intervention (1). AT-MSCs were injected in the parotid and submandibular glands in 10 patients with radiation-induced salivary gland hypofunction and xerostomia and stimulated whole saliva was collected by sialometry (2). Whole saliva samples from baseline (prior to the AT-MSC treatment), day 5 and 120 days after intervention were analyzed by nano-scale liquid chromatography tandem mass spectrometry to explore changes induced by the AT-MSCs (3). Created using Biorender (VR241Q1V5K).

REVIEWERS' COMMENTS:

Reviewer #2 (Remarks to the Author):

authors have satisfactorily addressed my prior comments. No additional suggestions.

Reviewer #3 (Remarks to the Author):

Thank you for your nice rebuttal. I have no further comments.